# Diagnostic and Prognostic Value of Angiogenic Status in Hereditary Hemorrhagic Telangiectasia

**DOI:** 10.3390/diagnostics14242783

**Published:** 2024-12-11

**Authors:** Sherlyne Jaimes-Díaz, Gustavo Juan-Samper, Susana Torres-Martínez, Eva Escorihuela-Alares, Silvia Calabuig-Fariñas, Raquel Rodríguez-López, Nieves Prieto-Colodrero, Mercedes Ramon-Capilla, Estrella Fernández-Fabrellas

**Affiliations:** 1Pneumology Department, General University Hospital of Valencia, 46014 Valencia, Spain; gustavo.juan@uv.es (G.J.-S.); mercedes.ramon@uv.es (M.R.-C.); esferfa@gmail.com (E.F.-F.); 2Molecular Oncology Laboratory, General University Hospital of Valencia, 46014 Valencia, Spain; susana.torres@alu.umh.es (S.T.-M.); escorihuela_eva@gva.es (E.E.-A.); silvia.calabuig@uv.es (S.C.-F.); 3Laboratory of Molecular Genetics, General University Hospital of Valencia, 46014 Valencia, Spain; rodriguez_raqlop@gva.es; 4HGUV Biobank, Research Foundation, General University Hospital of Valencia, 46014 Valencia, Spain; prieto_nie@gva.es

**Keywords:** plasmatic angiogenic biomarkers, clinical significance, hereditary hemorrhagic telangiectasia diagnosis

## Abstract

**Background/Objectives**: Angiogenesis is involved in the pathogenesis of hereditary hemorrhagic telangiectasia (HHT). VEGF, ANG2, TGFβ1, and ENG are the most studied angiogenic factors, but their clinical significance in blood samples is still not completely defined. The genetic study of HHT mutations is the test of choice for diagnosing the disease, but this route is expensive, and the causative mutation is not found in up to 10% of cases. Therefore, the use of angiogenic biomarkers could facilitate a cheaper and easier approach to the diagnosis of HHT. To determine the diagnostic and prognostic value of the VEGFA, TGFβ1, ANG2, and ENG plasmatic concentrations in patients with HHT. **Methods**: All the participants were clinically evaluated and the concentrations of these angiogenic factors were measured using MILLIPLEX^®^MAP immunoassays in plasma samples collected from 44 patients with HHT and 19 controls. To evaluate the diagnostic validity of these parameters, we estimated the maximum Youden index of the ROC curve and evaluated their diagnostic value using multiple logistic regression. **Results**: Patients with HHT had increased blood levels of TGFβ1 and decreased ENG compared to the control group. We could not identify any angiogenic markers related to the clinical severity or epistaxis. TGFβ1 and ENG exhibited a higher discriminant capacity for HHT, especially patients with HHT1, and it was possible to develop signatures of these factors with diagnostic value. **Conclusions**: We identified several angiogenic factors that may be important diagnostic biomarkers for HHT and propose that the combination of TGFβ1 and ENG could represent a signature with diagnostic value for this disease.

## 1. Introduction

Hereditary hemorrhagic telangiectasia (HHT), also known as Osler–Weber–Rendu disease, is a rare genetic disorder characterized by the development of abnormal blood vessels called arteriovenous malformations (AVMs) throughout the body. AVMs are formed when arteries and veins directly connect, bypassing the local capillary system. Smaller AVMs, known as telangiectasias, are prone to bleeds, which can lead to various complications [1]. Researchers have recently focused on identifying angiogenic markers that can help in the diagnosis and treatment of HHT [2]. Thus, this current article explores the diagnostic value of angiogenic markers in HHT.

Angiogenesis is the process of forming new blood vessels from existing ones and is crucial in the development and maintenance of tissues and organs. However, in conditions like HHT, angiogenesis becomes dysregulated, resulting in the formation of abnormal blood vessels. Understanding the molecular mechanisms underlying angiogenesis can provide valuable insights into the pathology of HHT. One important angiogenic marker is vascular endothelial growth factor (VEGF), a signaling protein that promotes blood vessel growth. VEGF levels are often elevated in patients with HHT, thereby contributing to the formation and enlargement of telangiectasias [3]. Furthermore, targeting VEGF or its receptors has shown promising results in reducing the severity and frequency of nosebleeds, a common symptom of HHT [4].

Another angiogenic marker of interest is endoglin (ENG), a protein involved in blood vessel development and maintenance [5]. Mutations in the ENG gene have been linked to HHT and decreased ENG levels have been observed in patients with HHT. Moreover, researchers have found that measuring ENG levels can help diagnose HHT as well as monitor disease progression [6,7]. Additionally, therapies targeting ENG signaling are being investigated as potential treatments for HHT [8,9]. Besides VEGF and ENG, other angiogenic markers, such as angiopoietin 2 (ANG2) and transforming growth factor beta 1 (TGFβ1), have also been implicated in HHT. Angiopoietins are growth factors that regulate blood vessel formation both in the embryonic and postnatal phases [10], while TGFβ is involved in multiple cellular processes, including angiogenesis [11]. Several groups have measured the levels of TGFβ1 in different contexts, as the prototypic TGFβ family member, finding that the dysregulation of these markers can contribute to the abnormal blood vessel development seen in HHT.

The identification of angiogenic markers for HHT has opened new avenues for the diagnosis, prognosis prediction, and treatment of the disease. Furthermore, biomarker- based tests could also help to identify individuals at risk of developing HHT and in monitoring the disease progression in affected individuals [12]. Thus, further research in this field will undoubtedly enhance our understanding of HHT and improve patient care. The genetic study of HHT mutations is the best means of diagnosing this disease, especially in patients with a family history of the disease or a known HHT mutation. Of note, to date, more than 1000 different ENG and ACVRL1 mutations have been identified in families with HHT. ENG Mutation Database URL: http://www.arup.utah.edu/database/ENG/ENG_welcome.php (accessed 15 February 2022). Acvrl1 Mutation Database URL: http://www.arup.utah.edu/database/ACVRL1/ACVRL1_welcome.php (accessed 15 January 2022). ENG encodes the TGFβ type I receptor endoglin, which is commonly expressed in endothelial cells (ECs) [13,14,15], while ACVRL1 encodes the TGFβ type I receptor activin receptor-like kinase I, which forms a complex with ENG to bind bone morphogenetic protein 9 (BMP-9), a regulator of angiogenesis [15,16].

Nevertheless, genetic diagnosis is expensive, and the disease-causing mutation is still not identified in up to 10% of cases [17]. Therefore, in patients with clinical manifestations that do not meet all the Curaçao criteria (the presence of epistaxis, telangiectasia, a family history of HHT, or organic involvement) or that meet them but a mutation that can explain the disease is not found, investigating angiogenic markers could represent a faster, easier, and cheaper approach to diagnosis. Moreover, some markers have also been related to the severity of clinical manifestations [18]. Given this context, the objective of this current study was to quantify components of the HHT signaling pathway, including VEGFA, TGFβ1, ANG2, and ENG, in patients with HHT and in controls, to analyze their diagnostic capacity and usefulness as disease severity markers in our population of patients with HHT.

## 2. Materials and Methods

### 2.1. Patients and Demographic Characteristics

Between January 2021 and December 2022, 44 patients from the Department of Pneumology in the General University Consortium Hospital of Valencia were included in this study. Of these, according to the Curaçao criteria, 38 had a definitive HHT diagnosis, 5 had a possible diagnosis, and 1 was an asymptomatic carrier. Nineteen unrelated volunteers with no history of known malignancy, trauma, surgery, or chronic medication were included as controls. Age and sex were no different between the patients and controls. In the controls, 37% were male, 63% were female, and 73% were in the highest age range of 40 to 77 years, while 27% were aged 10 to 40 years. Among the HHT patients, 43% were male, 57% were female, and 71% were in the highest age rank of 40 to 77 years, while 29% were aged 10 to 40 years. Patients with HHT were classified into two subgroups: HHT1 and HHT2, according to whether the ENG or ACVRL1 mutation was involved, respectively (Table 1).

All the patients were clinically evaluated and the complementary tests required according to the Clinical Guidelines for the Diagnosis and Management of Hereditary Hemorrhagic Telangiectasia [19] were applied. Diagnostic screening tests, including blood tests, brain magnetic resonance imaging, liver ultrasound imaging, gastroduodenoscopy, and/or colonoscopy, were performed in selected cases based on their clinical and analytical criteria for suspected gastrointestinal bleeding. Pulmonary arteriovenous malformation screening was also performed with transthoracic contrast echocardiography, extending the study with pulmonary computed tomography in cases positive for right-to-left intrapulmonary shunt. This study was approved by the General University Hospital of Valencia Consortium Ethics Committee, and the consent of all the participants was obtained prior to their inclusion in the study.

### 2.2. Determination of Angiogenic Factors by Immunoassays

Samples from patients included in this study were provided by the Biobank, Research Foundation University General Hospital of Valencia, which is integrated into the Valencian Biobank Network and the Instituto de Salud Carlos III Biobanks and BioModels platform, and they were processed following standard operating procedures. The sample consisted of peripheral venous blood from 44 patients with HHT and 19 healthy controls, which was collected in 10 mL EDTA tubes and centrifuged at 2000× *g* for 10 min. The blood plasma was stored at −80 °C until analyzed collectively for the study markers by a researcher blinded to the disease status of the participants.

Plasma concentrations of ENG, VEGFA, and ANG2 were determined using MILLIPLEX HAGP1MAG-12K-03, Angiogenesis/GF MAG 1 (part number HAGP1MAG-12K-03), and MILLIPLEX TGFβ1 Single Plex MAGNETIC Bead (part number TGFBMAG-64K-01) kits (Merck KGaA, Darmstadt, Germany). The plasma samples were diluted 1:3 in phosphate-buffered saline before use. All the procedures were conducted according to the manufacturer’s instructions, and all the samples were tested together with the quality controls and standard samples provided in the kit. After completing all the steps, the 96-well plate was analyzed using a Luminex^®^ 200 instrument with xPONENT^®^ v3.1 software (Luminex Corp, Austin, TX, USA), configuring the equipment as described in the manufacturer’s instructions. The results were expressed in picograms per milliliter (pg/mL) for the plasma levels and the serum levels of the angiogenic factors were assessed using absolute concentration values.

### 2.3. Statistical Analysis

Kolmogorov–Smirnov tests were employed to analyze the data, and, given the asymmetry of the distribution and presence of outliers, coupled with the limited sample size of some subgroups, a non-parametric analysis approach was used, reporting the results as medians and interquartile ranges (IQRs). Comparisons of each angiogenic factor between the patients with HHT and controls were performed using Mann–Whitney U tests with Bonferroni correction for pairwise tests. Correlations between the concentration of the angiogenic factors and the severity of organ involvement in patients with HHT were determined by calculating the Spearman rank coefficients. Indicators of the diagnostic validity, sensitivity, and specificity of the parameters were estimated based on the cut-off values from the maximum Youden index of the receiver operating characteristic (ROC) curve used to discriminate the patients and controls. The area under the curve (AUC) estimate, 95% confidence interval, and AUC = 0.5 contrast test results are also provided.

Finally, we used multiple binary logistic regressions to evaluate the diagnostic value of these factors in combination, obtaining several regression equations that allowed us to estimate the probability of a given patient having HHT.

## 3. Results

### 3.1. Demographic Characteristics

Described in Table 1.

### 3.2. Relationship Between Plasmatic Biomarker Levels and Diagnosis

Our results indicate that there was good homogeneity between all the distributions we compared (Table 2 and Figure 1).

For VEGFA, there were no differences between the control group and the patients with HHT (*p* = 0.225), nor between each pair of diagnostic subgroups (*p* > 0.05 in the three multiple comparisons). In turn, the median level of TGFβ1 in patients with HHT was significantly increased compared to healthy patients (*p* < 0.001), both in the case of HHT1 (*p* = 0.003) and HHT2 (*p* = 0.006), with respect to the controls. However, there were no differences in the TGFβ1 between the patients with HHT1 and HHT2 (*p* = 1.000). In patients with HHT, the ANG2 levels were lower (median = 1.309) than those in the controls (median = 1.651), with this result nearly reaching statistical significance (*p* = 0.065). Finally, the ENG levels were significantly lower (*p* = 0.001) in patients with HHT (median = 581.7) than in the controls (median = 1.205). Of note, the ENG levels were much lower in patients with HHT1 compared both to those with HHT2 (*p* < 0.001) and the controls (*p* < 0.001); there were no differences in the ENG levels between the controls and patients with HHT2 (*p* = 1.000). These results show that TGFβ1 and ENG could be differential markers in HHT diagnosis.

### 3.3. Relationship Between Plasmatic Biomarkers, Epistaxis, and Hereditary Hemorrhagic Telangiectasia Severity

None of the estimated correlations was significant in terms of the disease severity (epistaxis severity plus number of affected organs), although a certain weak correlation (r = 0.265) could be seen for TGFβ1 (*p* = 0.086). Similarly, there was no significant correlation between the severity of the epistaxis and any of the angiogenic markers. There was only a weak trend for TGFβ1 (*p* = 0.097), but the magnitude of this correlation was weak (r = 0.25) (Figure 2).

### 3.4. Diagnostic Validity of the Angiogenic Markers Studied

#### 3.4.1. Patients with Hereditary Hemorrhagic Telangiectasia and Controls

To discriminate between healthy controls and patients with HHT, ROC curves were estimated for each angiogenic marker (Figure 3), with these indicating (Table 3) that both the TGFβ1 and ENG parameters exhibited a higher discriminant capacity.

The cut-off point identified for TGFβ1 implies that this factor showed high sensitivity and moderate specificity. In turn, the cut-off point identified for ENG indicated that its specificity was high with fairly moderate sensitivity. Thus, a model was developed combining these two parameters, from which the regression equation *p*/1 − *p* = 3.71 × 1.000TGFβ1 × 0.99868ENG was obtained, where *p* is the probability of having HHT. This equation allowed for the probability of having HHT to be estimated with a sensitivity of 74.4% and a specificity of 84.3%, with the optimal cut-off point being an AUC of 0.8 for the ROC curve (Table 4).

#### 3.4.2. Hereditary Hemorrhagic Telangiectasia Type 1 and Controls

In the case of discriminating between the healthy controls and patients with HHT1, the ROC curves and diagnostic values of the markers studied were different. Of note, a sensitivity of 95.0% and specificity of 94.8% were achieved by determining the ENG values alone. By adding the TGFβ1 values (Table 5), a sensitivity of 95% and specificity of 100% were obtained when using the regression equation in the context of a combined model: *p*/1 − *p* = 3.71 × 1.00015TGFβ1 × 0.99868ENG, where *p* is the probability of having HHT1. Thus, this model improved the diagnostic capacity of these parameters because the AUC (0.96) for the ROC curve remained constant with respect to the individual model based on ENG only, while the specificity improved to 100%.

#### 3.4.3. Hereditary Hemorrhagic Telangiectasia Type 2 and Controls

For HHT2, we found no justification for the development of a combined model. The model based exclusively on TGFβ1 was the best possible option.

## 4. Discussion

Our main findings were that the plasmatic concentrations of TGFβ1 and ENG were different in patients with HHT and the controls, with elevated TGFβ1 levels and decreased ENG values in patients with HHT. Of note, ENG was only decreased in patients with HHT1. In addition, we were unable to find any markers of the disease severity or epistaxis, probably because of our small sample size. Thus, we found that analyzing the blood levels of TGFβ1 and ENG can contribute to the diagnosis of HHT, especially HHT1.

Several authors have detected increased VEGF concentrations in patients with HHT compared to controls [3,20,21], proposing VEGF as a possible HHT screening marker. Although we identified elevated levels of VEGF in patients with HHT compared to controls, the difference was not significant, also probably because of our sample size. Giordano et al. [22] also reported increased VEGF levels in patients with HHT and suggested that VEGF could be related to the disease severity, especially in terms of epistaxis. It might also be useful to study the VEGF levels before initiating anti-angiogenic treatments, especially Avastin (anti-VEGF antibodies), as a possible predictor of the treatment response [23]. We identified a weak, non-significant correlation between VEGF and the epistaxis severity, which was stronger for TGFβ1 (Figure 2).

The ANG2 levels also seemed to be reduced in patients with HHT when compared to the controls [3,17]; however, our data showed a non-significant trend in this respect. ANG2 is a proangiogenic factor strongly expressed in vasculature remodeling during embryogenesis, inflammation, and tumor-driven angiogenesis, which can also serve as a diagnostic marker for leukemia [24] and sepsis. Thus, it represents a good candidate as a predictor of the HHT severity for future studies.

TGFβ1 belongs to a large, transversal family of ligands that act on virtually all cell types [25], including ECs, with compelling evidence emerging from genetic, pharmacological, and histopathologic studies in this respect. Indeed, several groups have measured the TGFβ1 levels in patients with HHT. Sadick et al. [21] reported increased plasma concentrations of TGFβ1 in German patients with HHT compared to the controls, while Letarte et al. [26] found lower plasma levels of TGFβ1 in Canadian patients with HHT compared to the controls. These discrepancies may originate in the different genetic backgrounds of the populations involved. We found significantly increased TGFβ1 plasma concentrations compared to the controls, suggesting that, in the context of ENG haploinsufficiency, the TGFβ1-mediated inhibition of EC proliferation was impaired [23,24,25]. In addition, modifier genes regulating TGFβ1 expression may act in concert with HHT genes and could be associated with multiple phenotypes of the disease [2,21]. Nonetheless, TGFβ1 is not considered a reliable biomarker because there are multiple discrepancies in the academic literature regarding its use [27]. However, given our results, it seems reasonable to still include TGFβ1 as a diagnostic marker for HHT.

The ENG levels are decreased in patients with HHT1, which strongly supports haploinsufficiency as the underlying cause of the disease [28]. ENG is strongly expressed in vascular ECs, while its deficiency induces vascular injury in vivo and inhibits capillary tube formation in vitro [29]. Abdalla et al. [7] described how the ENG levels should be lower in patients with HHT1 compared to controls because of ENG haploinsufficiency. ENG deficiency is also detected in macrophages and ECs from HHT2 patients [30]. Given that the ENG levels are significantly different in patients with HHT1, HHT2, and controls [17], it could serve as an HHT biomarker that can distinguish HHT1 and HHT2. Our results are consistent with previous publications in relation to ENG [3,7,17,22] and suggest that HHT1 can be diagnosed with a sensitivity of 95% and specificity of 94.8% with ENG data alone, and that the addition of TGFβ1 determinations can increase the specificity to 100%. Thus, to improve the discriminating capacity demonstrated for the biomarkers in our series, we created a combined model that includes both ENG and TGFβ1 and verified that the joint model improved the discriminating capacity of the biomarkers for the overall diagnosis of HHT.

It is important to consider the risk of not diagnosing HHT in asymptomatic children and young adults in which the Curaçao criteria can fail and in whom visceral AVMs can have serious consequences. In these cases, not all the typical symptoms may be present because of age-related penetrance and the variable onset of the disease [31,32,33,34]. Thus, the genetic diagnosis of HHT in these cases remains a challenge. The molecular testing algorithm consists of a multigene panel that includes ACVRL1, ENG, GDF2, and SMAD4, and at our institution, multiplex ligation-dependent probe amplification was also incorporated into the panel in 2020. Since then, if no results are obtained after sequencing these four genes, we examine the exome using the extended HHT panel and according to the degree of clinical suspicion. This is a lengthy, very expensive process, and in 10% of cases, the mutation still remains unidentified in clinically affected patients [31,32] in whom other pathways related to vascular development are likely involved [5,6,7,8,9,10,11,12,13,14,15,16,17,18,19,20,21,22,23,24,25,26,27,28,29,30,31,32,33,34,35,36,37,38,39]. In this scenario, biochemical tests using plasma from these patients could quickly confirm a diagnosis when clinical symptoms appear.

Thus, in this context, the availability of a diagnostic model including validated biomarkers as a commercial kit would be more cost-effective compared to genetic tests and could facilitate the diagnostic process while awaiting massive genetic sequencing. Nevertheless, before the possible clinical application of the model we propose here, prospective, multi-site studies with larger cohorts of patients with a known genetic diagnosis of HHT will be required. Furthermore, the biomarkers included must also be standardized and their specificity and sensitivity determined, and the molecular technique and sample type used should also be defined. Previous studies have suggested associations between biomarkers and HHT [40] or the presence of AVMs in different, specific organs, while others have detected marker levels strongly associated with one phenotypic characteristic, thereby supporting the hypothesis that there are indeed circulating biomarkers associated with specific HHT phenotypes [41]. This highlights the need for further research on biomarker profiles associated with specific disease phenotypes and may even contribute to personalized risk–benefit analysis for HHT management.

HHT has an age-dependent penetrance and usually initially presents with recurrent epistaxis followed by the characteristic telangiectasias of the face, oropharynx, and hands over time. Although HHT exhibits complete penetrance after the age of 40, young patients may also present symptoms of the disease and are at risk of severe complications. The clinical presentation, and in particular the severity, increases with the patient age but is highly variable [42,43]. In our series, the individuals with more severe involvement were aged between 30 and 77 years, with an average age of 52 years. Although the direct descendants of HHT patients have a 50% chance of inheriting a mutation, some studies have found a higher prevalence of HHT in women compared to men. One possible explanation for this observation is that women consult with primary care providers significantly more frequently than men. Moreover, evidence of the influence of sex on the HHT severity is scarce and has not been directly addressed. Only data from the largest series of HHT patients that underwent liver transplantation, published by the European Liver Transplant registry, showed more severe liver involvement in woman than in men [44]. For this reason, the controls in this current study were selected to ensure that their age and sex were not significantly different from those of the patients, in order to avoid any bias.

The limitations of our study were (1) its small sample size; (2) the fact that no established reference cut-offs were established for any of the biomarkers; and (3) the biomarkers were measured in different clinical situations of HHT, which could have influenced their expression levels. In conclusion, progress in understanding the genetic basis, clinical characteristics, treatment, and diagnosis of HHT has recently been made but more detailed work on the different disease phenotypes is still required to improve its treatment and diagnostic instruments. Here, we highlight the value of angiogenic biomarkers in blood to aid the diagnosis of HHT and propose a signature of such factors (TGFβ1 and ENG) with diagnostic value [45,46].

## Figures and Tables

**Figure 1 diagnostics-14-02783-f001:**
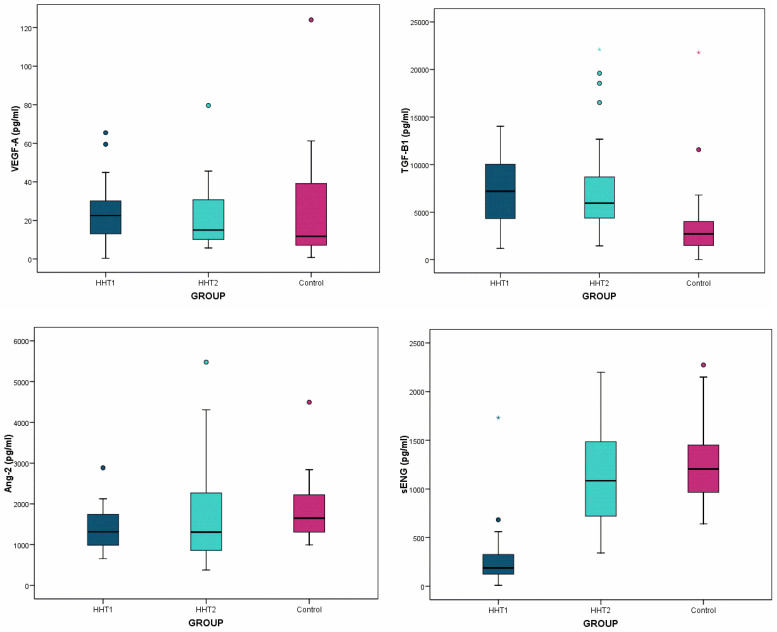
Box plots of the values of vascular endothelial growth factor (VEGFA), transforming growth factor beta 1 (TGFβ1), angiopoietin 2 (ANG2), and endoglin (ENG) in the studied population (patients with hereditary hemorrhagic telangiectasia type 1 or 2 and unaffected controls), with the boxes showing the median and interquartile ranges (25% and 75%) and the whiskers expressing the values in an acceptable range, with the outliers shown as circles and the extremes with asterisks.

**Figure 2 diagnostics-14-02783-f002:**
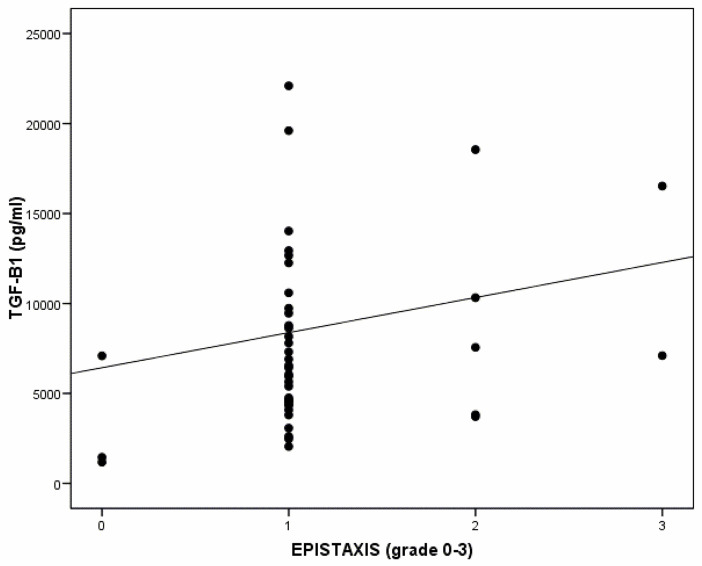
Graph relating the severity of epistaxis (epistaxis severity score (ESS): 0, not present; 1, mild, ≥1–≤4; 2, moderate, >4–≤7; 3, severe, >7–≤10) in patients with hereditary hemorrhagic telangiectasia against transforming growth factor beta 1 (TGFβ1) expression levels (*p* = 0.097).

**Figure 3 diagnostics-14-02783-f003:**
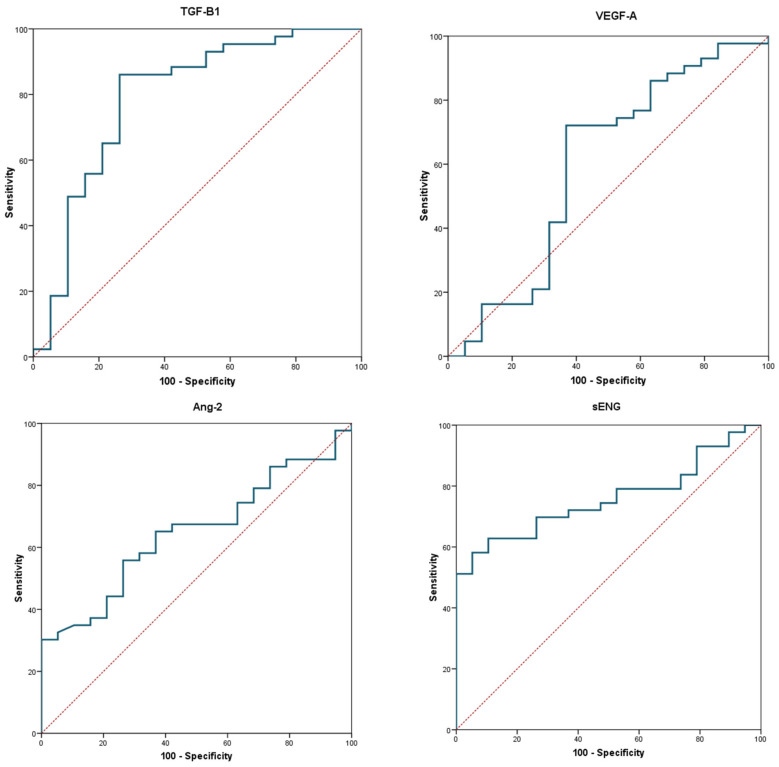
Receiver operating characteristic (ROC) curves for the transforming growth factor beta 1 (TGFβ1), vascular endothelial growth factor (VEGFA), angiopoietin 2 (ANG2), and endoglin (ENG) markers in patients with hereditary hemorrhagic telangiectasia.

**Table 1 diagnostics-14-02783-t001:** Patients and families with hereditary hemorrhagic telangiectasia type 1 and 2. (a) Patients with mutations in endoglin (ENG). (b) Patients with mutations in ACVRL1. (c) The age, sex and markers levels of the controls. Total of 44 patients with hereditary hemorrhagic telangiectasia (HHT) belonging to 18 families were included in the analyses of vascular endothelial growth factor (VEGFA), transforming growth factor beta 1 (TGFβ1), endoglin (ENG), and angiopoietin 2 (ANG2) plasma levels. Of these, 21 patients were genetically diagnosed as HHT1 because they had a mutation in ENG while the other 23 patients had HHT2 as the result of a mutation in ALK1. Severity (0–6 = Epistaxis severity score ESS: 0–3 + nº affected organs) Abbreviations: Ex, exon; In, intron; Pro, promotor; PAM, pulmonary arteriovenous malformation; E, epistaxis (ESS: 0, not present; 1, mild ≥1–≤4; 2, moderate >4–≤7; 3, severe >7–≤10) ; C, Cerebral arteriovenous malformation; G, Gastrointestinal arteriovenous malformation; L, Liver arteriovenous malformation; ABS, brain abscess; M, migraine; Ep, epilepsy; S, stroke; RAE, retinal artery embolism; Hp, hyperdynamic pulmonary hypertension; NI, not identified; CR, Clinical relevance; P, pathogenic; B, benign; USV, unknown significance variants. * rs753480401 clinical variant with no functional evidence or in silico prediction (ALAMUT V1.7.2) showing a discrepancy between predictors representative of a possible tolerated change. ** No previously described clinical variant data and no in silico prediction (ALAMUT V1.7.2) showing a discrepancy between predictors, representative of a possible tolerated change. It could not be established whether these changes were in the cis or trans conformation in these patients.

**(a) Patients and Families with THH1: Mutation in *ENG* Gene**												
**Family**	**Sex**	**Age**	**Severity**	**Clinic**	**VEGFA** **(pg/mL)**	**TGFβ1** **(pg/mL)**	**ANG2** **(pg/mL)**	**ENG** **(pg/mL)**	**Exon**	**Change**	**Type**	**CR**
**#01**	F	67	2	E1 P, M, RAE	24	6.910	1.543.3	301	In 1	c.68-2A>T	Splicing	P
**#01**	F	68	1	E1, NI	0.4	5.650	1.257.2	30	In 1	c.68-2A>T	Splicing	P
**#02**	M	47	3	E1, P, G, S	27	4.083	1.004.8	186	Ex 2	c.145G>T	Missense	P
**#02**	F	70	4	E2, P, G, S	43	7.564	1.309.1	9	Ex 2	c.145G>T	Missense	P
**#02**	M	44	2	E1, P	18	4.570	1.738	284	Ex 2	c.145G>T	Missense	P
**#02**	M	40	1	E1, NI	299	50.707	1.237	--	Ex 2	c.145G>T	Missense	P
**#02**	F	28	2	E1, P, Ep	66	9.461	1.159	39	Ex 2	c.145G>T	Missense	P
**#02**	M	25	2	E1, P	8	7.315	2.126	192	Ex 2	c.145G>T	Missense	P
**#03**	M	77	4	E2, P, G, S, ABS	27	3.715	1.750	139	Ex 5	c.657C>G	Missense	P
**#05**	F	64	0	E0, NI	16	1.179	653	561	Ex 2	c.207G>A	Missense	B
**#05**	M	37	0	E0, NI., E	25	7.093	817	352	Ex 2	c.207G>A	Missense	B
**#06**	M	59	2	E1, G	29	10.597	709	135	Ex 4 Ex 8	c.392C>T c.1024C>T	Missense Nonsense	B P
**#06**	F	27	2	E1, C, M	59	12.943	2.887	684	Ex 4 Ex 8	c.392C>T c.1024C>T	Missense Nonsense	B P
**#06**	F	47	2	E2, NI	32	10.326	1.321	250	Ex 4 Ex 8	c.392C>T c.1024C>T	Missense Nonsense	B P
**#06**	F	36	1	E1, NI	16	6.428	1.779	205	Ex 4 Ex 8	c.392C>T c.1024C>T	Missense Nonsense	B P
**#06**	F	21	2	E2, P	5	2.054	1.913	1732	Ex 4 Ex 8	c.392C>T c.1024C>T	Missense Nonsense	B P
**#09**	F	41	3	E1, P, L	7	8.166	1.558	119	Pro/Ex 1	Pro/Ex1	Deletion	P
**#09**	F	46	2	E1, P, M	14	3.800	1.413	130	Pro/Ex 1	Pro/Ex1	Deletion	P
**#14**	F	24	2	E1, P, S, M	45	1.4030	1.131	90	Ex 8	c.1024C>T	Nonsense	P
**#14**	M	52	1	E1, NI	21	9.740	913	469	Ex 8	c.1024C>T	Nonsense	P
**#15**	M	35	4	E1, P, G, C	12	1.2259	968	133	Ex 9	c.1202delA	Frameshift	P
**(b) Patients and Families with THH2: Mutation *ACVRL1* Gene**												
Family	Sex	Age	Severity	Clinic	VEGFA (pg/mL)	TGFβ1 (pg/mL)	ANG2 (pg/mL)	ENG (pg/mL)	Exon	Change	Type	CR
**#04**	M	53	2	E1, L	20.67	12.673	1.188	522	Ex 7	c.929T>C	Missense	P
**#06**	M	76	3	E1, L, Hp	8.92	3.821	1.051	2.142	Ex 7	c.968A>T	Missense	P
**#07**	F	49	3	E1, L, C	5.96	4.628	2.291	2.198	Ex 10	c.1436G>A	Missense	P
**#10**	F	54	3	E1, P, L	15.04	4.337	2.697	1.140	Ex 10	c.1436G>A	Missense	P
**#10**	M	57	1	E1, NI	9.48	4.615	1.177	342	Ex 10	c.1436G>A	Missense	P
**#10**	M	16	0	E0, NI	8.78	1.451	3.504	2.094	Ex 10	c.1436G>A	Missense	P
**#11**	M	69	5	E3, P, L	14.14	7.103	965	1.691	Ex 4	c.350delG	Frameshift	P
**#12**	M	76	3	E3, NI	30.29	16.526	376	368.04	Ex 10	c.1436G>C	Missense	P
**#12**	F	49	1	E1, NI	18	4.415	834	1.468	Ex 10	c.1436G>C	Missense	P
**#12**	M	48	1	E1, NI	20	4.760	1.315	644	Ex 10	c.1436G>C	Missense	P
**#13**	F	75	3	E1, G, L	40	2.2097	1.546	1.502	Ex 3	c.236_237del	Frameshift	P
**#13**	F	53	1	E1, NI	15	8.783	5.478	1.461	Ex 3	c.236_237del	Frameshift	P
**#13**	F	29	1	E1, NI	10	6.059	1.309	582	Ex 3	c.236_237del	Frameshift	P
**#13**	M	52	2	E1, P	80	1.9605	3.376	659	Ex 3	c.236_237del	Frameshift	P
**#13**	F	16	1	E1, NI	44	2.470	1.945	1.086	Ex 3	c.236_237del	Frameshift	P
**#13**	F	48	4	E1, P, G, L, M	11	5.947	4.309	832	Ex 3	c.236_237del	Frameshift	P
**#13**	F	28	1	E1, NI	46	7.805	2.247	1.255	Ex 3	c.236_237del	Frameshift	P
**#16**	F	67	2	E1, P	31	8.635	873	919	Ex 7	c.1031G>A	Missense	P
**#16**	F	30	4	E2, G, L, Hp	39	18.550	1.595	1.237	Ex 7	c.1031G>A	Missense	P
**#17**	M	57	1	E1, NI	29	5.399	540	783	Ex 9 Ex 10	c.1252G>A * c.1465C>G **	Missense Missense	USV USV
**#18**	F	48	1	E1, NI	10	6.553	845	952	Ex 8	c.1120C>T	Missense	P
**#18**	F	54	2	E1, L	6	2.608	788	1.602	Ex 8	c.1120C>T	Missense	P
**#18**	M	57	1	E1, NI	14	3.074	791	1.007	Ex 8	c.1120C>T	Missense	P
**(c) Controls (Age and Gender)**												
Number			Sex	Age	VEGFA (pg/mL)		TGFβ1 (pg/mL)		ANG2 (pg/mL)		ENG (pg/mL)	
**1**			F	62	2.1		1205		1425		1306	
**2**			F	75	41.6		2760		2358		2150	
**3**			M	65	5.8		2710		1399		1046	
**4**			F	57	11.8		4666		1214		885	
**5**			F	50	7.7		1793		1662		1476	
**6**			M	69	8.4		0		2836		1902	
**7**			M	75	6.6		382		1051		1424	
**8**			F	32	11.2		3358		2777		1304	
**9**			F	27	40.7		11,575		2842		1089	
**10**			M	29	0.7		898		1732		1371	
**11**			F	28	11.8		1996		1105		914	
**12**			F	55	124.0		21,781		1651		1922	
**13**			F	60	37.6		3343		1572		1092	
**14**			M	32	3.13		3388		1622		866	
**15**			M	67	22.1		943		1154		641	
**16**			M	48	12.2		1956		993		1017	
**17**			F	61	9.9		2434		4495		782	
**18**			F	60	41.6		6804		1866		1205	
**19**			F	70	61.2		6037		2088		2274	

**Table 2 diagnostics-14-02783-t002:** Results (means and interquartile ranges) from the Mann–Whitney U tests with Bonferroni correction used to compare transforming growth factor beta 1 (TGFβ1), vascular endothelial growth factor (VEGFA), angiopoietin 2 (ANG2), and endoglin (ENG) values in populations with hereditary hemorrhagic telangiectasia type 1 or 2 and unaffected controls. Abbreviations: IR, interquartile range; HHT, hereditary hemorrhagic telangiectasia.

Patients	(*n*)	VEFGA (pg/mL)	*p*-Value	Patients	(*n*)	ANG2 (pg/mL)	*p*-Value
		Median (IR)				Median (IR)	
Controls	19	12 (7–41)	0.225	Controls	19	1651 (1214–2358)	0.065
HHT Total	44	19 (11–31)		HHT Total	44	1309 (939–1846)	
Controls	19	12 (7–41)	0.705 (Controls vs. HHT1) 1.000 (Controls vs. HHT2) 1.000 (HHT1 vs. HHT2)	Controls	19	1651 (1214–2358)	0.195 (Controls vs. HHT1) 0.495 (Controls vs. HHT2) 1.000 (HHT1 vs. HHT1)
Cases HHT1	21	24 (14–32)		Cases HHT1	21	1309 (1005–1738)	
Cases HHT2	23	15 (10–31)		Cases HHT2	23	1309 (845–2291)	
Patients	(*n*)	TGFβ1 (pg/mL)	*p*-Value	Patients	(*n*)	ENG (pg/mL)	*p*-Value
		Median (IR)				Median (IR)	
Controls	19	2710 (1205–4666)	<0.001	Controls	19	1205 (914–1476)	<0.001
HHT Total	44	6731 (4376–10,033)		HHT Total	44	582 (192–1237)	
Controls	19	2710 (1205–4666)	0.003 Controls vs. HHT1 0.006 Controls vs. HHT2 1.000 (HHT1 vs. HHT1)	Controls	19	1205 (914–1476)	<0.001 Controls vs. HHT1 1.000 Controls vs. HHT2 <0.001 HHT1 vs. HHT1
Cases HHT1	21	7315 (4570–10,326)		Cases HHT1	21	189 (124–326)	
Cases HHT2	23	5947 (4337–8783)		Cases HHT2	23	1086 (659–1503)	

**Table 3 diagnostics-14-02783-t003:** Diagnostic validity of the markers (transforming growth factor beta 1 (TGFβ1), vascular endothelial growth factor (VEGFA), angiopoietin 2 (ANG2), and endoglin (ENG)) used to identify patients with a diagnosis of hereditary hemorrhagic telangiectasia compared to healthy controls, according to the area under the curve (AUC). The 95% confidence intervals (CIs), *p*-values, optimal cut-off point (according to the Youden criteria), sensitivity (S), and specificity (E) are also shown.

	AUC	Cut-Off	S	E
TGFbeta-1	0.79 (0.65–0.93); *p* < 0.001	3551.4	86.0%	73.7%
VEGF-A	0.60 (0.43–0.77); *p* = 0.225	12.3	72.1%	63.2%
ANG2	0.65 (0.51–0.79); *p* = 0.065	1359.6	44.2%	73.7%
ENG	0.76 (0.64–0.87); *p* = 0.001	848.9	62.8%	89.5%

**Table 4 diagnostics-14-02783-t004:** The combined diagnostic validity of the transforming growth factor beta 1 (TGFβ1) and endoglin (ENG) markers for identifying a positive diagnosis of hereditary hemorrhagic telangiectasia compared to healthy controls, according to the area under the curve (AUC). The 95% confidence intervals (CIs), *p*-values, optimal cut-off point (according to the Youden criteria), sensitivity (S), and specificity (E) are also shown.

	AUC	Cut-Off	S	E
TGFbeta-1 + ENG	0.80 (0.69–0.91); *p* < 0.001	0.68	74.4%	84.3%

**Table 5 diagnostics-14-02783-t005:** The combined diagnostic validity of the transforming growth factor beta 1 (TGFβ1) and endoglin (ENG) markers for identifying patients with a diagnosis of hereditary hemorrhagic telangiectasia type 1 compared to healthy controls, according to the area under the curve (AUC). The 95% confidence intervals (CIs), *p*-values, optimal cut-off point (according to the Youden criteria), sensitivity (S), and specificity (E) are also shown.

	AUC	Cut-Off	S	E
TGFbeta-1 + ENG	0.96 (0.88–1.00); *p* < 0.001	0.49	95.0%	100%

## Data Availability

The data presented in this study are available upon request from the corresponding author due to privacy and ethical reasons.

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
