# Peer review of "Diagnostic and Prognostic Value of Angiogenic Status in Hereditary Hemorrhagic Telangiectasia"

_diagnostics, 2024, doi:10.3390/diagnostics14242783_

Round 1

Reviewer 1 Report

Comments and Suggestions for Authors

The authors measured the  VEGF, ANG2, TGFb1 and ENG levels in plasma of HHT patients and controls. They concluded that the TGFb1 and ENG levels could be valuable biomarkers to diagnose HHT. 

I have some suggestions for authors.

1. The control group is too small for the study to get appropriate conclusions. The authors should include more controls.

2. The table1 , which describes the information of HHT1 and HHT2 patients should include the detected gene name of the mutation. 

3. As the clinical presentation of HHT patients may be associated with age, the authors should additionally compare the four biomarkers based on age.

Author Response

Thank you very much for the reviewer’s comments, questions, and suggestions, which will undoubtedly improve the text of this work. Please find attached the answers to your questions and comments

1.- The control group is too small for the study to get appropriate conclusions. The authors should include more controls.

Without a doubt, we appreciate the reviewer’s suggestion, but it was somewhat challenging to find suitable controls, as they were individuals who should have ages similar to the patients and, importantly, should not regularly use medication or have any underlying pathology. The statistical procedures were applied appropriately to the sample size.

2.- The table1 , which describes the information of HHT1 and HHT2 patients should include the detected gene name of the mutation. 

We have added the name of the mutated gene in Table 1: Endoglin (ENG) in the first table and activin receptor-like kinase-1 (ALK1) in the second. We have also included a table with the controls data (age and gender)

3.- As the clinical presentation of HHT patients may be associated with age, the authors should additionally compare the four biomarkers based on age.

Data from the controls regarding gender and age have been included. The age and gender were not significantly different between the controls and the patients, with age divided into two groups: those older and younger than 40 years, as this is the age at which the expression of the disease becomes manifest. A comment has also been added in the text regarding the significance of these factors and the potential bias.

“Nineteen unrelated volunteers with no history of known malignancy, trauma, surgery or chronic medication were included as controls. Age and gender were no different between patients and controls. Controls 37% males ,73 % in the highest age range 40 to 77 years and 27% in the range 10 to 40 years.). HHT patients 43% males, 71% in the highest age rank 40 to 77 years and 29% in the range 10 to 40 years”.

Additionally, a sentence has been included in the text along with the referenced sources.

“HHT has an age-dependent penetrance and usually initially presents with recurrent epistaxis followed by the characteristic telangiectasias of the face, oropharynx, and hands over time. Although HHT exhibits a complete penetrance after the age of 40, young subjects may also present symptoms of the disease and are at risk of severe complications. The clinical presentation, and in particular the severity, increases with the age of the patients, but it is highly variable (42,43,44). In our series, the patients with more severe involvement were aged between 30 and 77 years, with an average age of 52 years. The controls were selected to ensure that age and gender were not significantly different from those of the patients, in order to avoid any bias for this reason”

Reviewer 2 Report

Comments and Suggestions for Authors

The manuscript by Jaimes-Diaz and colleagues analyses the levels of four factors involved in the angiogenic and inflammatory process in the serum of 44 patients diagnosed with HHT and in that of 19 volunteers. The results of the study show that ENG levels correlated significantly with the diagnosis of HHT type 1 (HHT1) in particular, and with HHT in general. The association of ENG levels with TGFBeta1 levels allowed the authors to identify an algorithm that correlates and identifies patients diagnosed with HHT1 with a specificity of approximately 95% and a sensitivity of 100%.

Despite the limitations associated with the small number of patients enrolled, the results of the study confirmed on larger cohorts could have a very important impact on the diagnostic practice of HHT.

My suggestion is to discuss further other aspects of the study's weaknesses that could partly undermine the results:

1. Sex was not considered in this study. Is there any evidence that sex like age can modulate disease onset, severity and diagnosis?

2. The age of the patients did not correlate with the expected results despite being an important parameter in the onset of this disease.

3. The same for the comorbidity and/or the intake by these patients and controls of drugs that could significantly alter plasma levels of the measured analytes.

These aspects all need to be properly reported in the demographics table and discussed.

Author Response

Thank you very much for the reviewer’s comments, questions, and suggestions, which will undoubtedly improve the text of this work. Please find attached the answers to your questions and comments
1. Sex was not considered in this study. Is there any evidence that sex like age can modulate disease onset, severity and diagnosis?
Although the offspring of HHT patients have a 50% chance of inheriting a mutation, some studies have found a higher prevalence of HHT in women compared with men. One possible explanation for this observation is that women have significantly higher rates of consultation with primary care providers. Moreover, the evidence of gender influence in HHT severity is scarce and not directly addressed. Only data from the largest series of HHT patients with liver transplantation from the European Liver Transplant registry, showed a more severe liver involvement in woman than men (45). In our series of controls and cases, the genders were balanced ( controls 37% M and HTT patients 43%M), and it does not seem reasonable that they had any influence.
2. The age of the patients did not correlate with the expected results despite being an important parameter in the onset of this disease.
Data from the controls regarding gender and age have been included. The age and gender were not significantly different between the controls and the patients, with age divided into two groups: those older and younger than 40 years, as this is the age at which the expression of the disease becomes manifest. A comment has also been added in the text regarding the significance of these factors and the potential bias.
“Nineteen unrelated volunteers with no history of known malignancy, trauma, surgery or chronic medication were included as controls. Age and gender were no different between patients and controls. Controls 37% males ,73 % in the highest age range 40 to 77 years and 27% in the range 10 to 40 years.). HHT patients 43% males, 71% in the highest age rank 40 to 77 years and 29% in the range 10 to 40 years”.
Additionally, a sentence has been included in the text along with the referenced sources.
“HHT has an age-dependent penetrance and usually initially presents with recurrent epistaxis followed by the characteristic telangiectasias of the face, oropharynx, and hands over time. Although HHT exhibits a complete penetrance after the age of 40, young subjects may also present symptoms of the disease and are at risk of severe complications. The clinical presentation, and in particular the severity, increases with the age of the patients, but it is highly variable (42,43,44). In our series, the patients with more severe
2
involvement were aged between 30 and 77 years, with an average age of 52 years. The controls were selected to ensure that age and gender were not significantly different from those of the patients, in order to avoid any bias for this reason”
3.- The same for the comorbidity and/or the intake by these patients and controls of drugs that could significantly alter plasma levels of the measured analytes.
Control volunteers were subjects with no history of known malignancy, trauma, surgery or chronic medication, and only one of the HHT patient received medication related with the pathology (bevacizumab).

Round 2

Reviewer 1 Report

Comments and Suggestions for Authors

I have some suggestions for authors.

1.The title line of Table 1 and 2 should incorporate the units of measurement for VEGFA, TGFb1, ANG2 and ENG to enhance its presentation. The annotations for the abbreviations utilized in the table should be positioned beneath the tables.

2. The authors included the data on age and sex for both control and HHT groups, but only provided a description of the male data. It's necessary to also include information on females.

Author Response

Thank you very much for the reviewer’s comments, questions, and suggestions, which will undoubtedly improve the text of this work. Please find attached the answers to your questions and comments

1.- The title line of Table 1 and 2 should incorporate the units of measurement for VEGFA, TGFβ1, ANG2 and ENG to enhance its presentation. The annotations for the abbreviations utilized in the table should be positioned beneath the tables.

Following your instructions, we have incorporated the units of measurement for VEGFA, TGFβ1, ANG2 and ENG in the title line of Table 1 and 2, which undoubtedly improve the presentation. We have also placed the annotations of the abbreviations used below the table

  1. The authors included the data on age and sex for both control and HHT groups, but only provided a description of the male data. It's necessary to also include information on females.

            We have added and highlighted the data related to women in the text, and we apologize for the error.

Round 3

Reviewer 1 Report

Comments and Suggestions for Authors

I haven't observed any changes mentioned by the authors in the response letter.

Author Response

We sincerely apologize, but unfortunately, due to an error, we did not send the corrected version with the latest reviewer suggestions. Please ensure that the final text does indeed include their suggestions."

I haven't observed any changes mentioned by the authors in the response letter.

Thank you very much for the reviewer’s comments, questions, and suggestions, which will undoubtedly improve the text of this work. Please find attached the answers to your questions and comments

1.- The title line of Table 1 and 2 should incorporate the units of measurement for VEGFA, TGFβ1, ANG2 and ENG to enhance its presentation. The annotations for the abbreviations utilized in the table should be positioned beneath the tables.

Following your instructions, we have incorporated the units of measurement for VEGFA, TGFβ1, ANG2 and ENG, highlighted in red, in the title line of Table 1 and 2, which undoubtedly improve the presentation. We have also placed the annotations of the abbreviations used below the table.

2. The authors included the data on age and sex for both control and HHT groups, but only provided a description of the male data. It's necessary to also include information on females.

We have added and highlighted the data related to women in the text, and we apologize for the error.

Round 4

Reviewer 1 Report

Comments and Suggestions for Authors

The figures should be in high-resolution.

Author Response

Thank you very much for the reviewer’s comments, questions, and suggestions, which will undoubtedly improve the text of this work. Please find attached the answers to your questions and comments:
The figures should be in high-resolution.
Following your instructions, we have changed the format of the figures to high resolution, which will undoubtedly make them easier to read. Thank you for the suggestion.
Yours faithfully,
Sherlyne Vanessa Jaimes Díaz
Valencia, Spain December 2, 2024
